# Cardiopulmonary Functional Profiles in Cancer Therapy-Related Cardiac Dysfunction Among Patients with Non-Hodgkin Lymphoma

**DOI:** 10.3390/diagnostics15040417

**Published:** 2025-02-08

**Authors:** Daniela Bursacovschi, Valeriu Revenco, Maria Robu, Oleg Arnaut

**Affiliations:** 1Department of Heart Failure, Public Healthcare Institution—Institute of Cardiology, MD-2025 Chisinau, Moldova; 2Department of Cardiology, Nicolae Testemitanu State University of Medicine and Pharmacy, MD-2004 Chisinau, Moldova; valeriu.revenco@usmf.md; 3Department of Hematology, Nicolae Testemitanu State University of Medicine and Pharmacy, MD-2004 Chisinau, Moldova; maria.robu@usmf.md; 4Department of Human Physiology and Biophysics, Nicolae Testemitanu State University of Medicine and Pharmacy, MD-2004 Chisinau, Moldova; oleg.arnaut@usmf.md; 5Bioinformatics and Computational Medicine Laboratory, Nicolae Testemitanu State University of Medicine and Pharmacy, MD-2004 Chisinau, Moldova; 6National Cancer Registry, Public Healthcare Institution—Oncological Institute, MD-2025 Chisinau, Moldova

**Keywords:** cancer therapy–related cardiac dysfunction, cardiotoxicity, cardiopulmonary test, heart failure

## Abstract

**Background:** Non-Hodgkin lymphoma (NHL) is associated with significant cardiovascular risks due to treatment-related toxicities, including cancer therapy-related cardiac dysfunction (CTRCD). While multimodality imaging, particularly echocardiography, is pivotal in monitoring cardiac function, the prognostic role of cardiopulmonary exercise testing (CPET) in predicting CTRCD remains underexplored. **Methods:** The prospective study enrolled 127 NHL patients, 72 men (56.7%) and 55 women (43.3%), with a median age of 62 years (range 34–83 years). The patients were assessed before initiating antitumor treatment and at six months follow-up using echocardiography and cardiopulmonary exercise testing. **Results:** Asymptomatic CTRCD occurred in 14.2% of NHL patients at six months of treatment. Patients with CTRCD exhibited significantly lower median work rates, volume of oxygen (VO_2_) at the anaerobic threshold, and O_2_ consumption efficiency, reflecting compromised metabolic and functional performance. Baseline peak oxygen consumption (VO_2_ peak) positively correlated with left ventricle ejection fraction (LVEF) at six months, while VO₂ peak < 14 mL/kg/min was negatively associated with LVEF. **Conclusions:** Asymptomatic CTRCD was identified in 14.2% of NHL patients at six months, with lower work rates, VO_2_ at the anaerobic threshold, and O_2_ consumption efficiency, indicating impaired performance. Baseline peak oxygen consumption correlated positively with LVEF, highlighting CPET’s potential for early CTRCD risk assessment.

## 1. Introduction

Non-Hodgkin lymphoma (NHL) encompasses a wide range of conditions, spanning from slow-progressing forms to highly aggressive cancers [1]. The occurrence of NHL has been rising consistently in recent years. In 2020, approximately 545,000 new cases were reported globally, along with 260,000 related deaths [2]. In NHL, achieving long-term survival relies not only on effectively treating malignancy but also on addressing the risks linked to treatment-related toxicity, with a focus on cardiotoxicity [3]. A meta-analysis by Boyne and colleagues revealed that long-term survivors of Hodgkin lymphoma and non-Hodgkin lymphoma have a markedly elevated risk of cardiovascular-related mortality, being 7.3 and 5.35 times greater, respectively, than that of the general population [4].

Cardiovascular adverse events associated with NHL therapy typically encompass cancer therapy-related cardiac dysfunction (CTRCD), myocarditis, vascular toxicities, hypertension, cardiac arrhythmias, QT interval prolongation, and pericardial vascular heart disease. The link between treatment approaches, such as anti-cancer drugs or radiotherapy, and these adverse effects is either well-documented or remains under study [5]. For conditions such as cardiac injury, cardiomyopathy, and heart failure, the latest cardio-oncology guidelines and expert consensus advocate the use of the term cancer therapy-related cardiac dysfunction [6]. The incidence of CTRCD in this population varies based on factors such as treatment regimens and individual patient characteristics. A study focusing on lymphoma patients receiving anthracycline-based chemotherapy reported a CTRCD incidence of 17.4% [7]. Multimodality imaging, with a focus on echocardiography, plays a crucial role in assessing cardiac function both prior to and throughout antitumor treatment [8].

Cardiorespiratory fitness, as measured by a cardiopulmonary exercise test, reflects the integrated capacity of the cardiopulmonary system to deliver adequate oxygen and substrate to active skeletal muscles for adenosine triphosphate resynthesis. Cardiorespiratory fitness, therefore, provides a robust, objective evaluation of global physical functioning. While cardiopulmonary exercise testing (CPET) is not a primary diagnostic tool for cancer therapy–related cardiac dysfunction, its potential prognostic significance has been largely overlooked [9]. Greater cardiorespiratory fitness, typically measured by peak oxygen consumption (VO_2_ peak), has been linked to a reduced risk of cancer development [10] and improved survival rates in large groups of cancer patients [11]. A single-center study on adult-onset cancer patients, with a median follow-up of nearly five years, identified higher cardiorespiratory fitness as an independent predictor of overall mortality, cancer-specific mortality, and cardiovascular mortality [12]. Taking this into consideration, we carried out this study to identify specific CPET parameters that could serve as early indicators of CTRCD, enhance understanding of the underlying mechanisms, and potentially guide personalized interventions to mitigate cardiovascular risks in cancer patients.

## 2. Materials and Methods

The study was conducted prospectively thanks to a collaboration between the Public Medical-Sanitary Institution “Institute of Oncology” from which patients were randomly selected, and the Public Medical-Sanitary Institution “Institute of Cardiology” also from Chisinau, Republic of Moldova, where they were investigated. The group of patients included in the study consisted of those diagnosed with non-Hodgkin lymphoma, aged over 18 years, who provided informed consent. The inclusion of patients in the study took place between 2022 and 2024, after the study was approved by the Ethics Committee of the State University of Medicine and Pharmacy “Nicolae Testemitanu” (approval number 7/28 December 2021). The research was carried out in accordance with the Helsinki Declaration. Patients were excluded from the study if they had a history of another oncological condition, prior treatments such as chemotherapy, immunotherapy, radiotherapy, or pre-existing cardiac conditions, including known coronary or myocardial diseases (such as prior myocardial infarction, chronic ischemic syndromes, various cardiomyopathies, or post-inflammatory/infiltrative myocardial disorders). Additionally, those with moderate to severe valvular abnormalities (including moderate or greater valvular regurgitation and/or stenosis) or heart failure classified as class III-IV according to the New York Heart Association (NYHA) classification were not included.

During the study implementation, patients were asked through a questionnaire about cardiovascular disease risk factors (diabetes, dyslipidemia, smoking, and hypertension) and their comorbidities (chronic kidney disease, proteinuria, chronic pulmonary obstructive disease, and thyroid gland pathology). During the study periods, data about the characteristics of non-Hodgkin lymphoma and the treatment applied (groups of chemotherapeutic/immunotherapeutic agents, and total dose of doxorubicine at 6-month follow-up) were collected. The patients were evaluated twice: first, before the initiation of specific antitumor treatments, and the second time after 6 months of treatment. They were assessed through echocardiography, cardiopulmonary exercise testing, and serological tests. For the serological tests regarding the analysis presented here, we note that we assessed the quantitative levels of troponin I and NT-pro BNP at two stages: before the initiation of specific treatment and at 6 months after its start.

The echocardiographic examination was conducted by the same cardiologist using a General Electric Vivid E95 ultrasound machine (GE VINGMED ULTRASOUND A/S, Horten, Norway). The assessment of the cardiac chamber size and function was performed in accordance with the guidelines provided by the American Society of Echocardiography and the European Association of Cardiovascular Imaging for cardiac chamber quantification from 2015 [13]. As per the diagnostic criteria outlined in the ESC 2022 Cardio-Oncology Guidelines, patients were categorized into two groups: those with CTRCD (Group I) and those without CTRCD (Group II), based on the observed changes in left ventricular ejection fraction, NT—pro BNP/troponin I levels. Table 1 outlines the diagnostic criteria for asymptomatic CTRCD and the corresponding patient classification from the study.

Patients were introduced to the testing equipment of the cardiopulmonary test machine (Quark CPET Cosmed, Rome, Italy), including the cycle ergometer and ventilation assessment tools, and were instructed to perform at maximum effort while stopping the test if symptoms such as chest pain, palpitations, or dizziness occurred. Communication was conducted through non-verbal signals to avoid interfering with the accuracy of measured parameters. The testing setup included a cycle ergometer, a mask for collecting gas samples, and a system for analyzing the composition of exhaled gasses. Simultaneous monitoring included a 12-lead ECG, blood pressure, and oxygen saturation.

The testing protocol followed a graded exercise design, where the intensity of physical effort increased progressively each minute during the exercise phase. The CPET protocol comprised four stages: rest, unloaded pedaling, incremental exercise, and recovery. The initial phase, lasting 3 min, was used to record baseline parameters such as ECG, heart rate, oxygen saturation, volume of oxygen (VO_2_), and respiratory rate, while also allowing the patient to adapt to the equipment. This was followed by a 3 min phase of unloaded pedaling. The incremental phase involved gradually increasing the workload over 8–12 min. The ramp rate, adjusted by the investigator based on the patient’s clinical status, ranged from 5 to 25 Watts. The formula for calculating the incremental ramp during CPET is as follows:Ramp effort (W/min) = (VO_2_ peak − VO_2_ unloaded)/100
whereVO_2_ unloaded (mL/min) = 150 + (6 × body weight (kg))VO_2_ peak (mL/min) = (height (cm) − age (years)) × 20 (for men)VO_2_ peak (mL/min) = (height (cm) − age (years)) × 14 (for women)

The exercise stage continued until the patient either reached maximal effort or was unable to maintain a steady cadence. Patients were asked about any symptoms both during and after the test, and the Borg scale was used to evaluate their effort at the end. The recovery phase, lasting approximately 5 min, involved unloaded pedaling. Monitoring continued during this phase until the ECG normalized, blood pressure stabilized, and heart rate decreased by 10 beats per minute compared to baseline. The CPET parameters evaluated included hemodynamic metrics (initial heart rate, peak heart rate, initial and final systolic and diastolic blood pressure, and initial/final oxygen saturation), VO_2_ peak, VO_2_ at the anaerobic threshold, anaerobic threshold as a percentage of VO_2_, volume of oxygen per heart rate (VO_2_ pulse), volume of oxygen per work rate (VO_2_/WR), maximum respiratory exchange ratio, minute ventilation, ventilation efficiency (minute ventilation/VCO_2_), respiratory reserve, end-tidal CO_2_ pressure, and the oxygen consumption efficiency curve. Furthermore, we examined the presence of a VO_2_ peak < 14 mL/kg/min, indicative of reduced aerobic capacity, which is linked to a higher risk of cardiovascular events and mortality. The workflow chart of patient recruitment is presented in Figure 1.

### Statistical Analysis

The analysis of the primary data was conducted using RStudio (version 2024.09.1+394, available at https://www.rstudio.com/) and Python (version 3.12.3, accessible via https://www.python.org/). These platforms facilitated full reproducibility of the statistical procedures employed. Descriptive statistics for numerical variables included calculations for the minimum, maximum, mean, and standard deviation, as well as the median, interquartile range (IQR). To compare numerical variables across groups, the Mann–Whitney test was utilized. Box plots were used to visualize group differences for numerical variables, while correlation analysis (Spearman test with 95% confidence intervals) was represented via heatmaps. For categorical variables, absolute and relative frequencies were determined, with relative frequencies accompanied by their respective 95% confidence intervals. Hypothesis testing for categorical data employed Pearson’s Chi-squared test, specifically the Monte Carlo variant with 100,000 samples. Across all statistical analyses, the significance threshold (α) was maintained at 0.05.

## 3. Results

### 3.1. Clinical and Demographic Characteristics

The study enrolled 127 patients diagnosed with NHL, comprising 72 men (56.7%) and 55 women (43.3%), with a median age of 62 years (IQR = 14.0), ranging from 34 to 83 years old. Group I included 18 patients (14.2%) who developed CTRCD at 6 months follow-up, and the remaining patients were included in Group II (no CTRCD). Patients’ characteristics and comorbidities are presented in Table 2. It should be noted that patients with a body mass index (BMI) > 30 kg/m^2^ were statistically significantly more prevalent in Group I, with 8 subjects (44.4%, 95% CI: 21, 67), compared to 23 subjects in Group II (21.1%, 95% CI: 13, 29), as indicated by the Monte Carlo test result of 4.6, *p* = 0.043. Dyslipidemia was also more prevalent among patients with CTRCD, affecting 17 patients (94.4%, 95% CI: 84, 105) compared to 67 patients (71.5%, 95% CI: 52, 71) in Group II, with a Monte Carlo test result of 7.5, *p* = 0.006. Additionally, the severity of arterial hypertension was significantly higher among patients who subsequently developed CTRCD, indicating a greater degree of hypertension in this group, with a Monte Carlo test result of 24, *p* < 0.001.

The patients in Group I exclusively presented with B-cell NHL, while in Group II, 105 patients (93.3%; 95% CI: 93–100) had B-cell NHL, and 4 patients (3.7%; 95% CI: 0.14–7.2) were diagnosed with T-cell NHL. Regarding disease stage distribution, the majority of patients were classified as stage IV, accounting for 89 individuals (70.1%, CI: 62, 78). This was followed by 18 patients (14.2%, CI: 8.1, 20) in stage III, 16 patients (12.6%, CI: 6.8, 18) in stage II, and only 4 patients (3.1%, CI: 0.11, 6.2) in stage I. In terms of NHL treatment, most patients underwent a combination of chemotherapy and immunotherapy, totaling 114 cases (89.8%, CI: 84, 95). Chemotherapy alone was administered to 7 patients (5.5%, CI: 1.5, 9.5), while 5 patients (3.9%, CI: 0.55, 7.33) received immunotherapy as a standalone treatment. Doxorubicin was included in the treatment protocols for 80 patients (63.0%, CI: 55, 71), while 47 patients (37.0%, CI: 29, 45) did not receive anthracyclines. The median cumulative dose of doxorubicin at the end of the study was 250 mg/m^2^ (IQR = 410.0). The specifics of the oncological disease and treatment aspects are presented in Table 3.

At baseline, Group I had a median left ventricular ejection fraction (LVEF) of 57.5% (IQR = 5.3), while Group II had a median LVEF of 59.0% (IQR = 5.0). The difference was not statistically significant (Mann–Whitney test = 767, *p* = 0.14). At the 6-month follow-up, patients who developed CTRCD had a median LVEF of 49.0% (IQR = 6.5), compared to 56.0% (IQR = 3.5) in those who did not, showing a statistically significant difference (Mann–Whitney test = 175, *p* < 0.001).

The initial NT-proBNP levels were significantly different between the groups. Group I had a median NT-proBNP value of 76.5 pg/mL (IQR = 270.5), while Group II had a median value of 41.0 pg/mL (IQR = 32.0), with a Mann–Whitney test result of 1491 and *p* < 0.001. At the follow-up, NT-proBNP levels in Group I increased to 423.0 pg/mL (IQR = 601.8), compared to 69.0 pg/mL (IQR = 32.5) in Group II, with a statistically significant difference (Mann–Whitney test = 1434, *p* < 0.001).

### 3.2. Correlations Between Baseline Cardiopulmonary Parameters and Left Ventricular Function at 6-Month Follow-Up

All patients underwent echocardiographic evaluation, as previously mentioned, both at baseline and at the 6-month follow-up. About the left ventricle, an analysis of the dynamics of echocardiographic parameters during treatment revealed a statistically significant increase in end-diastolic volume, rising from a baseline value of 131.1 mL (SD = 17.7) to 143.1 mL (SD = 15.6) with a Mann–Whitney test yielded a result of 513, *p* < 0.001. The initial mean left ventricular ejection fraction was 59.4% (SD = 3.3), while at the 6-month follow-up, it had decreased to an average of 55.7% (SD = 4.3), a statistically significant difference as indicated by the Mann–Whitney test 596, *p* < 0.001.

During the CPET, not all patients included in the study were able to complete the test. In Group I, 1 patient was unable to perform the test, while in Group II, 3 patients were unable to complete it. The inability to perform CPET was attributed to the advanced stage of the LNH. Consequently, the CPET data analyzed included results from 17 patients in Group I and 106 patients in Group II. We conducted an analysis of the relationship between cardiopulmonary parameters and changes in left ventricular function over time. This is depicted in Figure 2, which features a heatmap illustrating Spearman correlations between various parameters from the initial CPET performed at the start of the study and the LVEF measured after 6 months of treatment. Among the analyzed parameters, VO_2_ peak showed the strongest positive correlation with LVEF at 6 months (0.27). This suggests that better functional performance during the baseline CPET may predict improved left ventricular function at the third follow-up visit. Conversely, a VO_2_ peak < 14 mL/kg/min (indicating reduced peak oxygen consumption below the threshold of 14 mL/kg/min) was negatively correlated with LVEF at 6 months (−0.39), highlighting that patients with very low baseline VO₂ peak are more likely to experience reduced ejection fractions after 6 months. The work rate (WR) exhibited a moderate positive correlation (0.24), indicating that higher exercise capacity at baseline is associated with improved LVEF after 6 months. Similarly, a respiratory exchange ratio (RER) ≥ 1 (respiratory exchange ratio of 1 or higher) demonstrated a moderate positive correlation (0.23) with LVEF at the final assessment, suggesting that achieving maximal effort, as indicated by RER ≥ 1, is linked to better LVEF outcomes. A weaker positive correlation (0.18) was identified with the oxygen uptake efficiency slope (OUES), indicating that patients with higher baseline OUES values tend to show better LVEF at 6 months.

### 3.3. Analysis of Cardiopulmonary Fitness and Its Impact on Patients with and Without CTRCD

The maximum heart rate achieved in patients from Group I had a median of 123 bpm (IQR = 11.0), while in Group II, the median was 124.0 bpm (IQR = 22.8). The external mechanical workload or power output generated during exercise showed a trend toward statistical significance, with Group I achieving a median work rate of 115.0 Watts (IQR = 23.0) and Group II achieving a median of 124.0 Watts (IQR = 38.8). This difference was evaluated using the Mann–Whitney test (U = 633, *p* = 0.05), Figure 3.

A maximal cardiopulmonary test, analyzed by assessing the respiratory exchange ratio, showed no difference between the two groups (Mann–Whitney test = 735, *p* = 0.2). Although no statistically significant differences were observed between the groups in terms of median RER values (Mann–Whitney test = 736, *p* = 0.2), with a median of 1.1 (IQR = 0.2) in Group I and 1.1 (IQR = 0.1) in Group II, a statistically significant difference was noted in the proportion of patients achieving an RER ≥ 1.10. A RER ≥ 1.10 was significantly more frequent among patients who did not develop CTRCD compared to those who did. In the group without CTRCD, an RER ≥ 1.10 was observed in 79 participants (74.5%, 95% CI: 66, 83) compared to 8 patients (47.1%, 95% CI: 23, 71) in the group that developed CTRCD. This difference was statistically significant (Monte Carlo test = 5.3, *p* = 0.026). Among cardiovascular parameters, for final systolic blood pressure, no significant differences were observed between the groups. In Group I, the median systolic blood pressure was 145.0 mmHg (IQR = 16.0), compared to 150.5 mmHg (IQR = 14.0) in Group II (Mann–Whitney test = 787, *p* = 0.4). However, the final diastolic blood pressure showed statistically significant differences, with a Mann–Whitney test value of 563, *p* = 0.013. In Group I, the median diastolic blood pressure was 75.0 mmHg (IQR = 15.0) compared to 86.5 mmHg (IQR = 10.0) in Group II. An additional cardiovascular measure, such as oxygen pulse (VO₂ pulse), was not different between the patients before the specific antitumoral treatment started. The median value for Group I was 10.2 (IQR = 3.8) and 12.1 (IQR = 3.8) for Group II; Mann–Whitney test = 776, *p* = 0.4.

Analyzing the differences in parameters assessing metabolic gas exchange we observed a statistically significant impact on VO_2_ at anaerobic threshold and O_2_ consumption efficiency curve between the investigated groups. For VO₂ at anaerobic threshold, the median value in Group I was 880.0 mL/min (IQR = 262.0), while in Group II it was 1132.0 mL/min (IQR = 394.5), with a Mann–Whitney test result of 591, *p* = 0.023 (Figure 4).

Regarding the O₂ consumption efficiency curve, the median value in the group that developed CTRCD was 1811.0 (IQR = 457.0), whereas, in the group that did not develop CTRCD at 6 months of treatment, it was 2145.5 (IQR = 659.8), with a Mann–Whitney test result of 612, *p* = 0.034.

In the analysis of metabolic parameters, no statistically significant differences were observed between the groups in terms of oxygen consumption and peak oxygen uptake. The median VO_2_ peak was 13.6 mL/kg/min (IQR = 3.7) in Group I and 15.5 mL/kg/min (IQR = 4.4) in Group II, with a Mann–Whitney test result of 697, *p* = 0.14. However, a VO₂ peak < 14 mL/kg/min was more frequently noted in Group I compared to Group II, with 9 patients (52.9%, 95% CI: 29, 77) in Group I and 27 patients (25.5%, 95% CI: 17, 34) in Group II. This difference reached statistical significance, as indicated by the Monte Carlo test (5.3, *p* = 0.027), data presented in Figure 5.

Another parameter of interest investigated was VO_2_ peak (% predicted), which is considered a key tool for diagnosing exercise intolerance and identifying subclinical heart failure [14]. This indicator did not differ significantly between the patients. In Group I, the median was 61.0% (IQR = 13.0), while in Group II, it was 64.0% (IQR = 19.0), with a Mann–Whitney test result of 768, *p* = 0.3. Other parameters of gas exchange also did not differ significantly, as presented in Table 4.

The parameters evaluating ventilatory performance, assessed through cardiopulmonary testing, showed no statistically significant differences between patients who developed CTRCD and those who did not. The ventilatory efficiency slope (VE/VCO_2_ slope), representing the relationship between minute ventilation (VE) and carbon dioxide production (VCO_2_) during exercise, showed a lack of statistical significance between the analyzed groups, in Group I median value was 32.7 (IQR = 5.9) and in Group II median value was 31.2 (IQR = 5.4); Mann–Whitney test = 994, *p* = 0.5. The end-tidal partial pressure of carbon dioxide (Pet CO_2_), a key indicator of alveolar gas exchange that reflects how effectively the lungs are ventilating CO₂ out of the body, also did not show differences between groups analyzed; Mann–Whitney test = 1068, *p* = 0.2.

## 4. Discussion

The findings of our study provide early insights into the relationship between cardiopulmonary parameters and the development of cancer therapy-related cardiac dysfunction in patients diagnosed with non-Hodgkin lymphoma. There is published data about cardiorespiratory fitness, assessed via cardiopulmonary exercise testing through measures like peak oxygen consumption [15] or metabolic equivalents [16], that indicate highly reliable predictors of cardiovascular health and longevity [17]. It also enhances risk stratification, aiding in better health outcomes and management decisions [18]. The relation between baseline VO_2_ peak and LVEF at 6 months, reinforces the role of VO_2_ peak as an indicator of cardiopulmonary fitness and its prognostic significance. Other studies also have shown the role of VO_2_ peak as an important marker of cardiovascular health in oncology patients. Study data show that a higher VO_2_ peak predicts better cardiac outcomes in cancer survivors [19], aligning with our study’s findings that patients with VO₂ peak < 14 mL/kg/min were more likely to experience reduced LVEF at 6 months follow-up. As the VO_2_ peak reflects the maximal oxygen utilization during exercise, which is heavily influenced by cardiac output and skeletal muscle oxygen extraction, a low baseline VO_2_ peak may indicate pre-existing cardiac insufficiency or reduced aerobic capacity, making the heart more susceptible to further stress from cardiotoxic treatments. Reduced cardiac reserve at baseline suggests that the myocardium is less capable of handling the oxidative stress and injury induced by chemotherapy [20].

The current research identified a significant difference in the proportion of patients achieving a respiratory exchange ratio ≥ 1.10, with fewer patients with CTRCD reaching this threshold (*p* = 0.026). This validates the fact that maximal effort during CPET is an indicator of a better cardiac reserve, as mentioned by Jones et al. [21]. However, the absence of significant differences in median RER values between samples emphasizes additional research for a better understanding of the relationship between RER and CTRCD onset. Workload reflects the efficiency of the cardiovascular, respiratory, and muscular systems in sustaining physical effort.

Pre-treatment reductions in workload may indicate early signs of subclinical cardiac dysfunction, poor myocardial contractility, or reduced peripheral muscle efficiency, often related to systemic inflammation or comorbid conditions [22]. The lower median work rate suggests that it may be quantified as a predictor of vulnerability to cancer therapy-related cardiac dysfunction occurrence. This finding highlights the value of baseline-measured physical performance in identifying individuals at higher risk of CTRCD, ensuring earlier monitoring and preventive strategies. Other studies showed that reduced cardiorespiratory fitness is a major determinant of cardiovascular morbidity and cancer survival, suggesting that maintaining or improving cardiorespiratory fitness could be beneficial in mitigating treatment-related cardiac dysfunction [23].

VO_2_ at the anaerobic reflects the point at which oxygen delivery to muscles becomes insufficient, and anaerobic metabolism is required to meet energy demands. Patients with poor oxygen delivery mechanisms at baseline may be more vulnerable to chemotherapy-induced microvascular and mitochondrial damage, accelerating the onset of cardiac dysfunction [24]. The significant differences in work rate and VO_2_ at the anaerobic threshold between groups suggest a reduced exercise capacity in patients who developed CTRCD. These data correspond with reports from Lakoski et al., who showed that exercise intolerance correlates with subclinical cardiac dysfunction in cancer patients [25]. Furthermore, the lack of significant differences in metabolic parameters, such as the ventilatory efficiency slope, aligns with other resources, indicating that gas exchange efficiency may be less sensitive to CTRCD. Additionally, similar studies conducted by Arena et al. similarly noted that VE/VCO_2_ may have limited predictive value in early-stage cardiac dysfunction [26]. However, some published data demonstrate that a VE/VCO_2_ slope greater than 35 was associated with increased mortality after lung cancer resection, underlining its prognostic value in advanced disease stages [27].

Another point to discuss is that NHL can directly cause myocardial injury through several mechanisms. Direct myocardial infiltration by lymphoma cells leads to structural damage, reported in 10–20% of cases in autopsy studies, though often clinically silent [28]. Compression of cardiac structures by mediastinal or pericardial tumors can restrict filling or cause outflow obstruction [29]. Pericardial involvement, including effusion and constrictive pericarditis, further compromises heart function [30]. In the present study, we did not observe any clinical or investigative findings that would indicate the presence of these aspects, although the study duration was relatively short, with a follow-up period of six months.

### Limitations

One of the study’s limitations is the potential interaction effects between different chemotherapeutic regimens. Due to the limited sample size, it was not feasible to account for all possible combinations of regimens, which may influence the outcomes. Addressing this limitation would require a considerable increase in sample size to ensure adequate statistical power for detecting and analyzing such interactions. Future research should aim to include larger cohorts and consider stratified or interaction analyses to better elucidate the combined effects of these regimens. Furthermore, it is possible that the influence of cardiovascular risk factors on the development of CTRCD may have been underestimated, potentially leading to an incomplete understanding of their role in the progression of the condition.

The absence of a comparison between healthy controls and non-CTRCD patients is a potential limitation of this study, highlighting the need for future research to include such comparisons to provide a more comprehensive understanding of the full spectrum of cardiac effects associated with cancer therapy.

## 5. Conclusions

The findings point out the importance of baseline cardiopulmonary exercise in identifying patients at risk for cancer therapy-related cardiac dysfunction. Parameters such as VO_2_ peak, workload, VO_2_ at anaerobic threshold, and RER ≥ 1.10 are critical indicators of future cardiac performance and should be closely monitored for early high-risk individuals’ identification. These parameters can help stratify patients into risk categories, highlighting those who may benefit most from close monitoring during cancer treatment. Also, they could influence the choice or adjustment of cancer therapies, especially in patients with pre-existing low cardiac fitness. Regular CPET assessments can provide a framework for evaluating the effectiveness of interventions, ensuring that patients retain optimal cardiac function post-treatment.

Patients with a low VO_2_ peak or suboptimal exercise capacity might become candidates for targeted interventions, such as tailored exercise programs or cardioprotective strategies, to preserve cardiac function during and after cancer treatment. The results reinforce the role of CPET as a tool in cardio-oncology for improving long-term cardiac outcomes.

## Figures and Tables

**Figure 1 diagnostics-15-00417-f001:**
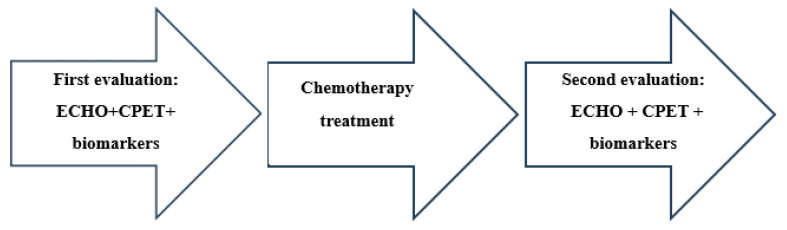
Workflow chart of patient recruitment.

**Figure 2 diagnostics-15-00417-f002:**
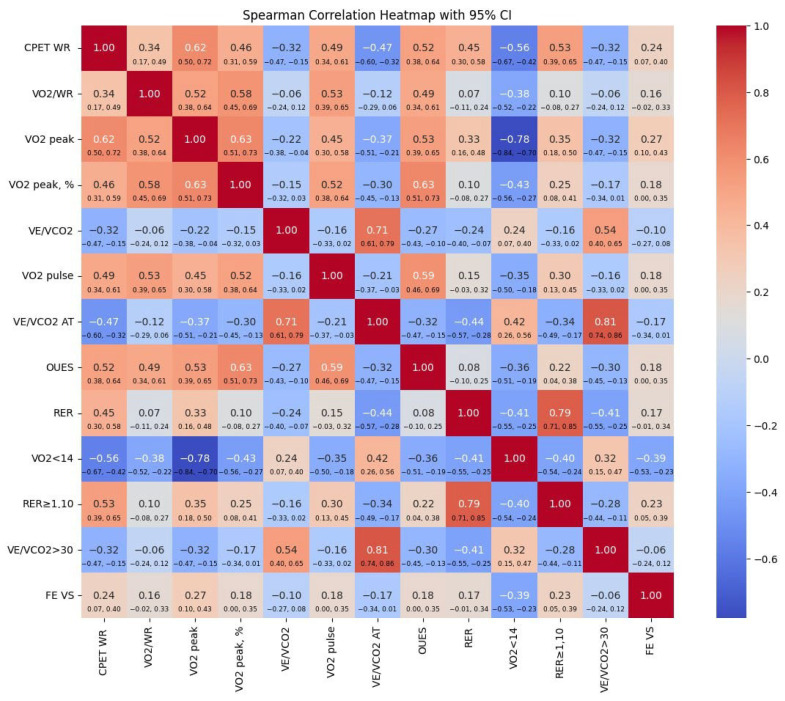
Heatmap of Spearman correlations linking initial cardiopulmonary parameters to left ventricular ejection fraction at 6-month follow-up. Note: CPET WR—work rate, VO_2_/WR—volume of oxygen per work rate, VO_2_ peak—peak oxygen uptake, VO_2_ peak, %—predicted peak oxygen uptake, VE/VCO_2_—ventilatory equivalent for carbon dioxide, VO_2_ pulse—volume of oxygen per heart rate, VE/VCO_2_ AT—ventilatory equivalent for carbon dioxide at anaerobic threshold, OUES—oxygen uptake efficiency slope, RER—respiratory exchange ratio, VO_2_ < 14—peak oxygen consumption < 14 mL/kg/min, RER ≥ 1.10—respiratory exchange ratio ≥ 1.10, VE/VCO_2_ > 30—ventilatory equivalent for carbon dioxide (VE/VCO_2_) exceeds 30 during exercise, and FE VS—left ventricular ejection fraction.

**Figure 3 diagnostics-15-00417-f003:**
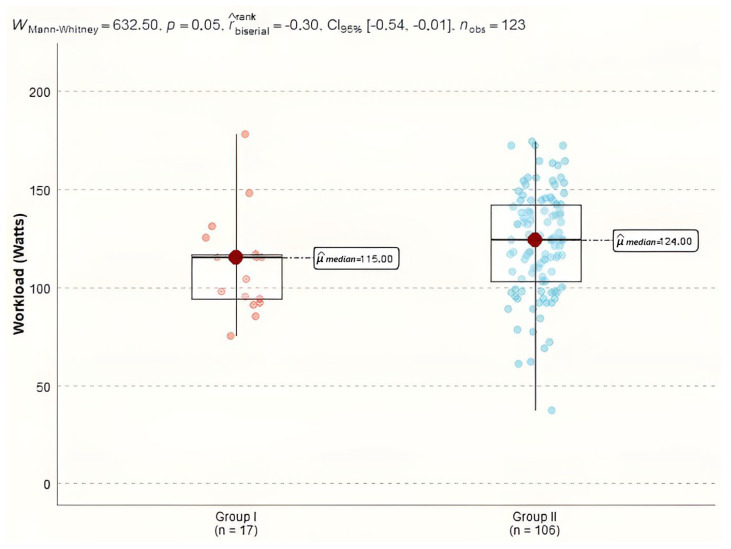
CPET workload in study groups.

**Figure 4 diagnostics-15-00417-f004:**
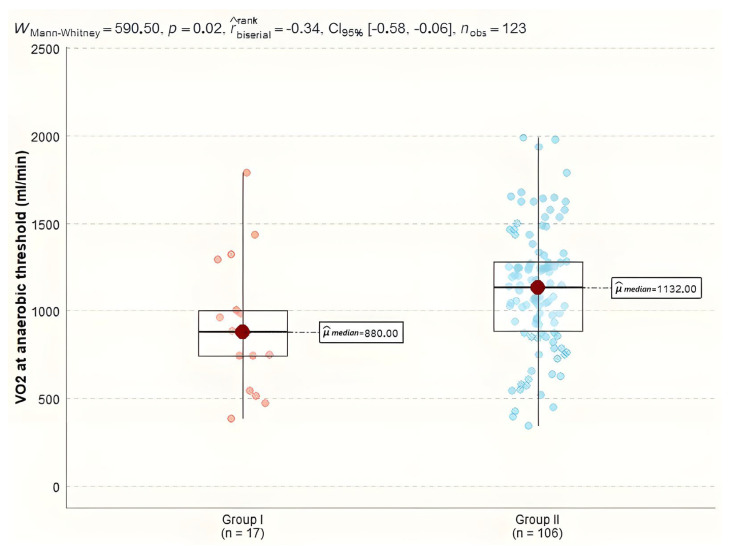
VO_2_ at anaerobic threshold in study groups.

**Figure 5 diagnostics-15-00417-f005:**
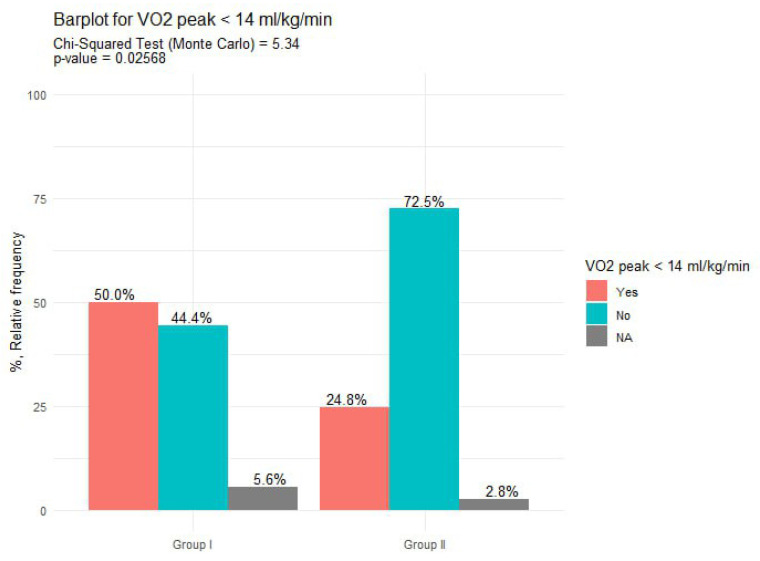
The distribution of patients with a VO_2_ peak < 14 mL/kg/min in study groups. Note: NA, Not Available.

**Table 1 diagnostics-15-00417-t001:** Definition criteria of asymptomatic CTRCD and distribution of patients.

Grade	Criteria	Number of Patients
Severe	New LVEF reduction < 40%	0
Moderate	New LVEF reduction by ≥10% to LVEF of 40–49% New LVEF reduction by <10% to LVEF of 40–49% AND new rise in cardiac biomakers ^1^	2 8
Mild	LVEF ≥ 50% AND new rise in cardiac biomarkers ^1^	8

^1^ cTnI > 99th percentile, NT-proBNP ≥ 125 pg/mL or new rise from baseline beyond the biological and analytical variation in the assay.

**Table 2 diagnostics-15-00417-t002:** Characteristics of patients with and without CTRCD.

	Group I (CTRCD, *n* = 18) ^1^	CI ^2^ 95%	Group II (No CTRCD, *n* = 109) ^1^	CI ^2^ 95%	Statistic Test	*p* ^3^
Age (years)	65.3 (11.9) 69.5 (16.5) 35.0 83.0	59, 71	60.9 (10.0) 62.0 (13.0) 34.0 80.0	59, 63	1243	0.071
Gender					1.3	0.3
Masculine	8 (44.4%)	21, 67	64 (58.7%)	49, 68		
Feminine	10 (55.6%)	33, 79	45 (41.3%)	32, 51		
Body mass index, kg/m^2^	28.2 (5.1) 28.7 (8.3) 18.6 36.8	26, 31	26.6 (5.6) 25.2 (6.6) 19.2 48.7	26, 28	1217	0.10
Body mass index > 30 kg/m^2^	8 (44.4%)	21, 67	23 (21.1%)	13, 29	4.6	**0.043**
Stroke history	0 (0.0%)	0.0	1 (0.9%)	−0.87, 2.7	0.17	>0.9
Peripheral arterial disease	0 (0.0%)	0.0	2 (1.8%)	−0.68, 4.4	0.34	>0.9
Smoking	4 (22.2)	3.0, 41	26 (23.9%)	16, 32	0.02	>0.9
Hypertension grade					24	**<0.001**
Grade I	2 (11.1%)	−3.4, 26	2 (1.8%)	−0.68, 4.4		
Grade II	6 (33.3%)	12, 55	29 (26.6%)	18, 35		
Grade III	6 (33.3%)	12, 55	5 (4.6%)	0.66, 8.5		
Diabetes Mellitus	5 (27.8%)	7.1, 48	13 (11.9%)	5.8, 18	3.2	0.13
Dyslipidemia	17 (94.4%)	84, 105	67 (61.5%)	52, 71	7.5	**0.006**
Chronic Kidney Disease					9.0	**0.029**
K/DOQI grade I/II	2 (11.1%)	−3.4, 26	12 (11.0%)	5.1, 17		
K/DOQI grade III	3 (16.7%)	−0.55, 34	2 (1.8%)	−0.68, 4.4		
Chronic obstructive pulmonary disease	5 (27.8%)	7.1, 48	22 (20.2%)	13, 28	0.53	0.5
Thyroid pathology	1 (5.6%)	−5.0, 16	8 (7.3%)	2.4, 12	0.07	>0.9

Note: ^1^ Mean (SD), median (IQR), minimum–maximum; *n* (%), ^2^ CI = confidence interval; ^3^ Mann–Whitney test; Pearson’s Chi-squared test with simulated *p*-value.

**Table 3 diagnostics-15-00417-t003:** Treatment regimens and non-Hodgkin lymphoma characteristics.

	Group I (CTRCD, *n* = 18) ^1^	CI ^2^ 95%	Group II (No CTRCD, *n* = 109) ^1^	CI ^2^ 95%	Statistic Test	*p* ^3^
**NHL type**					2.5	0.12
Indolent NHL	8 (44.4)	21, 67	70 (64.2)	55, 73		
Aggressive NHL	10 (55.6)	33, 79	39 (35.8)	27, 45		
**NHL stage**					4.3	0.2
NHL Stage I	0 (0.0)	0, 0	4 (3.7)	0.14, 7.2		
NHL Stage II	3 (16.7)	−0.55, 34	13 (11.9)	5.8, 18		
NHL Stage III	5 (27.8)	7.1, 48	13 (11.9)	5.8, 18		
NHL Stage IV	10 (55.6)	33, 79	79 (72.5)	64, 81		
**NHL by Primary Localization**					9.3	0.3
Abdominal	0 (0.0)	0, 0	8 (7.3)	2.4, 12		
Gastric	2 (11.1)	−3.4, 26	6 (5.5)	1.2, 9.8		
Hepatic	1 (5.6)	−5.0, 16	1 (0.9)	−0.87, 2.7		
Splenic	3 (16.7)	−0.55, 34	8 (7.3)	2.4, 12		
Retroperitonial	0 (0)	0, 0	1 (0.9)	−0.87, 2.7		
Mediastinal	1 (5.6)	−5.0, 16	1 (0.9)	−0.87, 2.7		
Multifocal	7 (38.9)	16, 61	41 (37.6)	29, 47		
Peripheral	4 (22.2)	3.0, 41	42 (38.5)	29, 48		
Thyroid	0 (0)	0, 0	1 (0.9)	−0.87, 2.7		
**Chemotherapeutic regimen**					3.9	0.4
R-CHOP	13 (72.2)	52, 93	55 (50.5)	41, 60		
CHOP	0 (0.0)	0, 0	5 (4.6)	0.66, 8.5		
R-COP	3 (16.7)	−0.55, 34	20 (18.3)	11, 26		
COP	0 (0.0)	0, 0	5 (4.6)	0.66, 8.5		
Other regimens	2 (11.1)	−3.4, 26	24 (22.0)	14, 30		
Doxorubicin administration	13 (72.2)	52, 93	67 (61.5)	52, 71	0.77	0.4
Ciclofosfan dosage ≥ 150 mg/kg	2 (11.1)	−3.4, 26	14 (12.8)	6.6, 19	0.04	>0.9
Vincristine administration	18 (100.0)	100, 100	104 (95.4)	91, 99	0.86	0.6
Rituximab administration	18 (100.0)	100, 100	97 (89.0)	83, 95	2.2	0.2

Note: ^1^
*n* (%), ^2^ CI = Confidence interval, ^3^ Mann–Whitney test; Pearson’s Chi-squared test with simulated *p*-value.

**Table 4 diagnostics-15-00417-t004:** The distribution of parameters assessing metabolic gas exchange between the study groups.

Parameters of Metabolic Gas Exchange	Group I (CTRCD, *n* = 18) ^1^	CI ^2^ 95%	Group II (No CTRCD, *n* = 109) ^1^	CI ^2^ 95%	Statistic Test	*p* ^3^
VO_2_ initial, mL/kg/min	5.5 (0.9) 5.3 (1.1)	5.1, 6.0	5.7 (1.3) 5.4 (1.7)	5.4, 5.9	863	0.8
VO_2_ peak, mL/kg/min	14.9 (2.8) 13.6 (3.7)	14, 16	16.1 (3.4) 15.5 (4.4)	15, 17	697	0.14
VO_2_ peak, (% predicted)	59.5 (12.8) 61.0 (13.0)	53, 66	63.4 (13.4) 64.0 (19.0)	61, 66	768	0.3
VO₂ at anaerobic threshold, mL/min	916.1 (372.5) 880.0 (262.0)	725, 1108	1120.1 (350.4) 1132.0 (394.5)	1053, 1188	591	**0.023**
VO_2_/WR, mL/min/Watt	8.1 (1.2) 8.3 (0.9)	7.5, 8.7	8.3 (1.3) 8.4 (1.8)	8.0, 8.5	791	0.4
Anaerobic threshold, % VO_2_	43.5 (15.8) 38.0 (25.0)	35, 52	50.0 (15.5) 52.0 (20.8)	47, 53	708	0.2
O_2_ consumption efficiency curve	1874.8 (411.2) 1811.0 (457.0)	1663, 2086	2103.4 (461.4) 2145.5 (659.8)	2015, 2192	612	**0.034**

Note: ^1^ Mean (SD), median (IQR); *n* (%), ^2^ CI = confidence interval, ^3^ Mann–Whitney test; Pearson’s Chi-squared test with simulated *p*-value.

## Data Availability

The data presented in this study are available on request from the corresponding author.

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
