# Peer review of "Cardiopulmonary Functional Profiles in Cancer Therapy-Related Cardiac Dysfunction Among Patients with Non-Hodgkin Lymphoma"

_diagnostics, 2025, doi:10.3390/diagnostics15040417_

Round 1

Reviewer 1 Report

Comments and Suggestions for Authors

This study was aimed to predict CTRCD in patients with NHL before initiating anti-tumor therapy and at six months follow-up using echo and CPET. This topic is of great clinical significance, especially in the field of chemotherapy-induced cardiotoxicity, as it provides valuable insights that can guide the implementation of intervention procedures. Understanding the relationship between specific CPET parameters and cardiac dysfunction can help identify patients at risk for heart failure or other cardiovascular complications early. However there are still some aspects needed to be improved in the current study:

Please provide the definition of asymptomatic CTRCD.

Please report which test was used to check the normality of data.

How to rule out the interaction effects between the different chemotherapeutic regimens, it would be nice to discuss on this aspect.

For asymptomatic patients with cardiac dysfunction, LVEF often doesn't show a significant decline in the early stage. It would be valuable to include additional echocardiographic parameters in the Tables and to compare the two groups with and without CTRCD, e.g. myocardial strain alterations. Exploring the relationship between myocardial deformation and changes in CPET parameters would also be interesting.

To determine whether myocardial injury is caused by NHL itself or results from anti-tumor treatment, more discussion needs to be considered.

To discuss the potential impact of the underlying disease, a comparison between the healthy control group and non-CTRCD patients at baseline should be conducted.

Please clarify more information for the role of CPET-derived predictors in subsequent intervention measures.

In the discussion section, it would be important to explain the potential mechanisms behind the observed changes in specific CPET parameters and their possible relationship with cardiac function alterations.

Some minor comments:

1. Please clarify more demographic information on this patient population in the abstract.

2. Would be nice to supplement the workflow chart of patient recruitment.

3. More lab test results related to the myocardial enzyme profiles should be presented in Table 1.

4. Please improve the resolution of Figures, and clarify the comparison between the two groups and the information each represents.

5. Given the relatively small sample size, it’s important to demonstrate the stability of the results and address the possibility of a lack of statistical significance due to the small sample size.

Author Response

the file in word

Reviewer 2 Report

Comments and Suggestions for Authors

The authors evaluated the significance of cardiopulmonary exercise testing (CPET) in determining the risk of developing asymptomatic cardiac dysfunction after a 6-month course of antitumor therapy in patients with lymphoma.

 Reviewer's comments

 Lines 159-160. The authors write, ' Descriptive statistics for numerical variables included ..... median and the 25th and 75th percentiles’. This is not in the text of the article.

Further, 'To compare numerical variables across groups, the Mann-Whitney-Wilcoxon test was utilized’. That is not quite correct, as far as I know, 2 tests are used in different cases, the Mann-Whitney and Wilcoxon tests. 

Tables 1 and 3. What is the point of different data presentation (Mean (SD), Median (IQR), Minimum - Maximum )? In my opinion, median and 25th and 75th percentiles (or mean+/-SD for normally distributed data), and p value are sufficient. What is the purpose of the Confidence Interval and Statistic Test? The same applies to the data given in the text of the article. 

The authors studied the relationship between initial cardiopulmonary parameters and left ventricular ejection fraction at 6-month follow-up. The authors should have provided ejection fraction values, baseline and at 6 months. 

The authors write “The external mechanical workload or power output generated during exercise was a statistically significant difference between the groups. Group I had a median work rate of 115.0 Watts (IQR=23.0), while Group II achieved a median of 124.0 Watts (IQR=38.8). This difference was confirmed by a Mann-Whitney test result of 633, p=0.049 (Figure 2)”. Figure 2 indicates p=0.05.; strictly speaking, there is no statistically significant difference here.

 In the Methods section the authors write ‘Patients were categorised into two groups: those with CTRCD (Group I) and those without CTRCD (Group II), based on the observed changes in left ventricular ejection fraction, NT - pro-BNP/troponin I levels’.  However, the text of the article does not provide the relevant data, baseline and after 6 months.  This is unacceptable. 

Patients have not been sufficiently characterised. Non-Hodgkin's lymphoma is a heterogeneous disease. The authors cite only the stage of the disease in the article.

 The study's additional limitations should be outlined. The number of patients in the groups is not comparable. The groups of patients with developed cardiac dysfunction and those with non-developed cardiac dysfunction differed at baseline in several parameters, including known cardiovascular disease risk factors. Could these factors have influenced the 6-month outcome? 

The paragraph after Figure 1 (lines 271-273) should be removed - these are someone else's comments.

 Figs are of low quality.

Author Response

.

Round 2

Reviewer 1 Report

Comments and Suggestions for Authors

Authors answered to all the comments and provided the changes necessary to improve the manuscript.

Author Response

Thank you for you meticulous work in reviewing our manuscript and for you valuable insights. We greatly appreciate the time and effort you have dedicated. We sincerely appreciate your contribution to improving the qaulity of our article.

Reviewer 2 Report

Comments and Suggestions for Authors

The authors have taken the main comments into account and made appropriate changes to the text of the article.

Regarding comment #3. I still do not agree with the authors that presenting data in all possible ways is correct. Moreover, the authors themselves write in their response to the reviewer: "Mean (SD) is useful for normally distributed data, and median (IQR) is useful for non-normally distributed variables". But this is not fundamental. If the editor finds it acceptable, this section can be left uncorrected.

There is a typo on the name of the test (line 170 - the Mann-Whitne).

Author Response

We sincereley appreciate your thorough review and the thoughtful suggestions you provided. Thank you for your constructive feedback, wich has significantly contributed to improving the clarity and quality of our work.

  • Regarding comment 3 - Our approach to presenting the data in this manner was driven by our commitment to transparency with the reader and our aim to provide the most detailed response possible for the analyzed data.  If the editor shares our perspective, we a re fully open to maintaining this presentation format.
  • The typographical error on line 170 was corrected.